# The Products of Probiotic Bacteria Effectively Treat Persistent *Enterococcus faecalis* Biofilms

**DOI:** 10.3390/pharmaceutics14040751

**Published:** 2022-03-30

**Authors:** Shatha Safadi, Harsh Maan, Ilana Kolodkin-Gal, Igor Tsesis, Eyal Rosen

**Affiliations:** 1Department of Endodontics, Goldschleger School of Dental Medicine, Sackler Faculty of Medicine, Tel-Aviv University, Tel-Aviv 699780l, Israel; shatha.safadi@gmail.com; 2Department of Molecular Genetics, Weizmann Institute of Science, Rehovot 76100001, Israel; harsh.maan@weizmann.ac.il; 3Department of Plant Pathology and Microbiology, The Hebrew University of Jerusalem, Rehovot 76100001, Israel; 4Center for Nanoscience and Nanotechnology, Tel Aviv University, Tel Aviv 6997801, Israel

**Keywords:** *Enterococcus faecalis*, biofilms, persistent infection, therapy, oral biofilms

## Abstract

**Objectives**: *Enterococcus faecalis* is a Gram-positive commensal bacterium that possesses various survival and virulence factors, including the ability to compete with other microorganisms, invade dentinal tubules, and resist nutritional deprivation. *E. faecalis* is associated with persistent endodontic infections where biofilms formed by this bacterium in the root canal frequently resist dental therapies. Aseptic techniques, such as the inclusion of sodium hypochlorite, are the most commonly used methods to treat *E. faecalis* infections within the root canal system. In this work, we assess the effectiveness of probiotic strains to prevent the regrowth of *E. faecalis* biofilm cells treated by sodium hypochlorite irrigation. **Methods:** First, methods are presented that evaluate the effects of short-term exposure to sodium-hypochlorite on established *E. faecalis.* Next, we evaluate the effects of the secreted products of probiotic strains on biofilm cells and planktonic cells. **Results:** Sodium hypochlorite, the treatment conventionally used to decontaminate infected root canal systems, was extremely toxic to planktonic bacteria but did not fully eradicate biofilm cells. Furthermore, low concentrations of sodium hypochlorite induced eDNA dependent biofilms. Strikingly, conditioned medium from the probiotic bacteria *Lactobacillus plantarum* and *Lactobacillus casei* was sufficient to fully prevent the regrowth of treated biofilms while showing reduced potency towards planktonic cells. **Conclusion:** Sodium hypochlorite irrigations may contribute to the persistence of biofilm cells if used at concentrations lower than 3%. Probiotic strains and their products represent a new reservoir of biofilm therapies for *E. faecalis* infections formed in the root canal system.

## 1. Introduction

Biofilms adhere to surfaces and interfaces [1], offering their microbial residents enhanced protection from environmental insults [2,3]. Microbial biofilms represent most of the microbial infections in the human body [4,5,6,7,8,9] and are particularly challenging in medical settings due to their inherent resistance to antimicrobial agents [10,11]. Bacteria within biofilm communities are 10-fold to 1000-fold more resistant to antimicrobial agents and antibiotics than planktonic (free-living) bacteria and are able to evade the immune system effectively [4,6,8,12,13].

*Enterococci* are human commensals that reside in various niches, such as the oral cavity and the gastrointestinal tract. *Enterococci* communities are resistant to a wide range of harsh conditions and therefore can persist in human hosts [14,15,16]. *Enterococcus faecalis* (*E. faecalis)* is a Gram-positive and facultative anaerobe and a prominent opportunistic pathogen. This bacterium is associated with diverse infections including but not limited to urinary tract infections, bacteremia, meningitis, infections of wounds, device-related infections [16,17,18] and antibiotic-resistant infections [19]. The emergence of drug-resistant strains that are retained by horizontal gene transfer the enzymes that degrade antibiotics, efflux pumps, and/or other resistance alleles, further reduces our capacity to manage these infections [20]. In addition, *E. faecalis* is capable of forming robust biofilms [21]. Biofilm growth is associated with an increased level of mutations and the induction of tolerance and virulence genes [13]. The pathways induced during biofilm formation include genes for surface adhesion, aggregation, extracellular polymers and toxins. These gene products can enhance biofilm formation and colonization as well as adherence to the host tissues. In addition, some *Enterococcus* strains are highly competent in the horizontal gene transfer of genes associated with virulence, antibiotic resistance, and tolerance [16,17,22].

In this context, we have recently reported [10] that *E. faecalis* grown on human dentin cannot be eliminated with sodium hypochlorite (NaOCl) solution. Importantly, *E. faecalis* persistent biofilm infections of the root canal system [10,23] are one of the most common antimicrobial-resistance-related diseases [19] and are also considered to be an appropriate model for studying the molecular mechanisms of biofilm resistance and testing novel antimicrobial treatments [24,25,26,27]. In these persistent infections, the infected root canal system is either necrotic or previously treated. Importantly, the immune system cannot access the root canal due to poor or lack of blood supply to the infected region. The necrosis of the tissue, as well as the withdrawal of the immune cells, contribute to the re-establishment of the infection following treatment [19,20,23].

As *E. faecalis* bacterial biofilms pose a major challenge in many biological systems and clinical scenarios, there is an urgent need to develop novel methods to overcome these resistant infections efficiently [19,26,28,29,30,31].

The current therapeutic approach to overcome resistant *E. Faecalis* biofilm root canal infections relies on common antimicrobial agents aimed to eradicate the bacteria. However, attempts to eradicate the infection from the root canal by different types of disinfectants and antimicrobial agents fail to eradicate the persistent biofilm. Furthermore, sodium hypochlorite (NaOCl), currently the most commonly used antimicrobial agent, has a limited ability to eradicate the biofilm, sometimes resulting in persistent infections [10,23]. This stresses the need to develop novel antimicrobial biofilm agents and novel comprehensive therapeutic approaches in order to overcome resistant *E. faecalis* biofilm infections. One of the requirements of an effective therapeutic approach is to be able not only to eradicate the biofilm within the infected root canal system during the root canal treatment but also to provide a long-lasting protective effect against the re-growth of the biofilm [23].

Probiotic microorganisms are living microorganisms that are capable of providing various health benefits to the host, including the elimination of microbial pathogens. Therefore, delivery of probiotics may eradicate or diminish more pathogenic bacteria [32] but is not currently used to treat root canal infections effectively. Species of the family Lactobacillaceae and Bacillus that are naturally found in the human gastrointestinal (GI) tract are often considered models of beneficial microbiota. These strains produce an extremely broad repertoire of antimicrobial organic acids, bacteriocins, and enzymes which were reviewed extensively [32]. Here, we demonstrate that specific probiotic strains *Lactobacillus casei* [33] and *Lactobacillus plantarum* [34] are superior to sodium hypochlorite irrigations in the elimination of established biofilms. Therefore, probiotic bacteria are capable of fully eradicating established infections within the root canal system as well as preventing re-infection following the root canal treatment. Thus, probiotics represent a novel and feasible therapeutic approach for these persistent infections.

## 2. Materials and Methods

### 2.1. Bacterial Strains and Growth Media

*Enterococcus faecalis* 29212, *Lactobacillus casei*, *Lactobacillus plantarum* and *Bacillus coagulans* ATCC 10545 strains were used in all experiments. *E. faecalis* and *B. coagulans* were grown in LB broth (Difco). *L. casei* and *L. plantarum* were grown in MRS broth. Assays were carried out with 50% TSB broth (Difco) enriched with 1% glucose (Sigma Aldrich, Waltham, MA, USA).

### 2.2. Conditioned Medium (CM) Preparation

A single colony of Lactobacillcae isolated from an MRS plate was grown overnight at 37 °C without shaking. Then, a diluted culture (1:20) was obtained by diluting the grown bacteria in 20 mL TSB glucose medium and the cells were further grown without shaking at 37 °C for 24 h.

A single colony of *Bacillus coagulans* isolated from the LB plate was grown overnight at 37 °C with shaking. Then, a diluted culture (1:20) was obtained by diluting the grown bacteria in a 20 mL TSB glucose medium, and cells were further grown with shaking at 37 °C for 24 h.

Cells were centrifuged and collected by filtration using a 0.22 μm filter (Corning, Corning, NY, USA). A fresh CM was prepared for each independent experiment.

### 2.3. Biofilm Formation and Disruption Assays

A single colony isolated from LB agar plates was grown in LB medium for 4 h at 37 °C with shaking. For biofilm growth, 1 μL of starter culture was diluted (1:100) in 100 μL TSB glucose media in 96-well polystyrene plates and incubated in 96-well polystyrene plates for 24 h. For inhibition assay, the growth media were applied with different concentrations of NaOCl (1%, 3% or 5%). Sodium hypochlorite solution was prepared using a stock solution of 6% sodium hypochlorite solution Sigma Aldrich, Waltham, MA, USA).

Biofilm formation was assessed by crystal violet staining. For disruption assay, NaOCl was applied for 5 min to the pre-established biofilm, and the biofilm remnant was assessed by crystal violet microtiter dish biofilm formation assay. In short, planktonic cells were removed by pipetting, and the adherent cells were stained with 0.1% crystal violet stain Sigma Aldrich, Waltham, MA, USA) for 15 min. The stain was removed, and the wells were washed with DDW (Deuterium-depleted water). Crystal violet was extracted with 95% ethanol and its intensity was determined by a spectrophotometer (OD 595_nm_).

### 2.4. Enzymes (DNase and Trypsin) Assays

*E. faecalis* biofilm was grown and treated with different concentrations of NaOCl as previously mentioned. After treatment, cells were washed with PBS and applied with PBS either applied or not with 50 µg/mL/500 µg/mL DNase/Trypsin. Treated biofilms were further incubated for 24 h at 37 °C. Crystal violet staining was performed for biofilm biomass assessment as described above.

### 2.5. Growth Measurements

A single colony isolated from LB agar plates was grown in LB medium for 4 h at 37 °C with shaking. Then, cells were grown in 300 μL of different concentrations (5%, 10%, 20%, 30% or 50%) of conditioned medium from either *L. casei*, *L. plantarum*, or *B. coagulans* in a 96-well microplate (Thermo Scientific, St. Louis, MO, USA), at 37 °C for 24 h, in a microplate reader. The optical density at 600 nm measurements was taken every 30 min.

Alternatively, cells of *E. faecalis* biofilm were harvested with PBS by harsh pipetting and grown in 300 μL of different concentrations (5%, 10%, 20%, 30% or 50%) of conditioned medium from either *L. casei*, *L. plantarum*, or *B. coagulans* in a 96-well microplate (Thermo Scientific, St. Louis, MO, USA), at 37 °C for 24 h, in a microplate reader. The optical density at 600 nm measurements was taken every 30 min.

### 2.6. Regrowth Measurements

A single colony isolated from LB agar plates was grown in LB medium for 4 h at 37 °C with shaking. Cells were diluted 1:25 into a fresh TSB glucose medium. 1000 μL of the culture were added to each well in a 12-well polystyrene plate and further incubated at 37 °C for 24 h. Biofilm was untreated or treated with different concentrations of NaOCl (1%, 3% or 5%) for 5 min. Bacterial cells were collected with 200 μL PBS by harsh pipetting and added to 3 mL of TSB glucose medium either mixed 1:1 with conditioned medium of *L. casei* as indicated in the legend for each figure. Cells were regrown at 37 °C with shaking. OD600 was measured manually in different time points (0 h, 4 h, 8 h, 16 h, 20 h, 24 h and 36 h)

### 2.7. Statistical Analysis

All experiments were performed five separate and independent times in triplicate, unless mentioned otherwise. Error bars represented ± SD, unless stated otherwise. *p* < 0.05 was considered statistically significant. Brown-Forsythe and Welch ANOVA tests with Dunnett’s T3 multiple comparison tests were used to compare data in order to correct for groups with significantly unequal variances. Data points at a particular interval in growth curves were compared using two-way ANOVA followed by Dunnett’s multiple comparison test. GraphPad Prism 9.0 (GraphPad Software, Inc., San Diego, CA, USA) was used to perform statistical analyses.

## 3. Results

During root canal treatments, the infected root canals are frequently irrigated with sodium hypochlorite with the overall aim of eliminating the pathogenic bacterial communities that are associated with the infection site completely. However, the concentration of sodium hypochlorite can differ regionally. In Europe, the sodium hypochlorite irrigation solution is 1%, while in the United States it is traditionally 3–5% [35]. We compared the response to different sodium hypochlorite concentrations as judged by two complementary parameters: the inhibition of biofilm formation, more compatible with a preventive scenario, and the destruction/killing of pre-established biofilms. In clinical scenarios, *E. faecalis* biofilms are already established at the root canal [23] and differences between the outcomes of both scenarios are of importance for formalizing the treatment protocol.

During an inhibitory scenario, biofilms were inhibited strongly by all clinically relevant sodium hypochlorite concentrations (Figure 1A), although no clear dose response was observed. However, in a clinically relevant scenario, where biofilms are formed prior to being treated, the 1% sodium hypochlorite solution treatment significantly improved biofilm adhesion as judged by crystal violet stain (Figure 1B). While the 3% and 5% sodium hypochlorite solutions reduced the biofilm biomass, they failed to eliminate biofilms altogether and residual biomass remained (Figure 1B).

### 3.1. eDNA Dependent Effect of Sodium Hypochlorite on the Biomass of Pre-Established Biofilms

As sodium hypochlorite irrigation was performed for a short time period (5 min) as compatible with clinical scenarios, the enhancement of associated biofilm biomass could not result from microbial replication (notably, *E. faecalis* division time is 30 min during planktonic growth). Therefore, we tested the hypothesis that extracellular polymers (proteins or eDNA) released during the partial cell lysis of the biofilm cells are responsible for the increased adhesion. As shown (Figure 2), DNase treatment but not proteinase K efficiently reduced the sodium-hypochlorite biomass to the untreated levels. As untreated biofilm was comparable to sodium hypochlorite induced biofilms treated by DNase, it is most likely that sodium hypochlorite increases the levels of eDNA by cell lysis, and this charged polymer later serves as an adhesion for *E. faecalis* biofilms. Proteinase treatment induced the biofilm biomass, potentially by increasing the availability of amino acids from the TSB media. However, as this was independent of sodium hypochlorite treatment, it has little relevance to the treatment protocol in *E. faecalis* biofilms.

### 3.2. The Effect of Sodium Hypochlorite Concentration on the Regrowth of Treated Biofilm Cells

Once biofilms are treated, they can potentially regrow once conditions become favorable in the oral cavity, and the ultimate goal in sodium hypochlorite irrigation is to reduce regrowth. To assess the effects of different concentrations of sodium hypochlorite on regrowth, we treated pre-established biofilms at 1–5% sodium hypochlorite solution, removed it, and added a fresh growth medium (Figure 3). The lag phase for biofilm regrowth was increased in all treated biofilms and was elongated in a manner proportional to the concertation of the sodium hypochlorite. No dose response was observed in the different sodium hypochlorite concentrations in the final carrying capacity (the OD reached by bacteria at the end of the logarithmic growth). However, in all working concentrations, re-growth was apparent and comparable to untreated control within 8 h. The application of sodium pyruvate (SP) (to mimic the glycolysis of glucose) as a carbon source had a minor rescue effect in low sodium hypochlorite concentrations but had little or no effect on the overall outcome of treatment as judged by the final carrying capacity of the cultures (Figure 3).

### 3.3. The Secretome of Probiotic Strains Prevents the Regrowth of Biofilm Cells

Recently, the use of probiotic bacteria for dental interventions was raised as a potential solution for root-canal infections [36]. To determine whether probiotic bacterial products can overcome *E. faecalis* pre-established biofilms, we treated the biofilms with the conditioned medium (CM) of *L. casei* following the removal of sodium hypochlorite. As shown, under all scenarios (Figure 4) the conditioned medium of *L. casei* prevented the regrowth of *E. faecalis* biofilms. To determine whether the conditioned media is a stand-alone treatment or synergistic with sodium hypochlorite, the untreated biofilms were treated with a conditioned media of the phylogenetically distinct member of the Firmicutes phylum, *Bacillus coagulans* (Appendix A). Unlike *L. casei*, the secreted products of *B. coagulans* were insufficient to prevent the regrowth of *E. faecalis* biofilms treated with sodium hypochlorite (Appendix A).

### 3.4. Comparing the Anti-Biofilm and Antimicrobial Effects of Probiotic Strains

To better comprehend the activities of the conditioned media from probiotic strains, we evaluated the bioactivity of the conditioned medium from three species, the bioactive *L. casei*, an additional Lactobacillaceae, *L. plantarum*, and the phylogenetically distant *firmicutes B. coagulans* on the untreated biofilms and planktonic cells of *E. faecalis.*

As shown, pre-established biofilms were sensitive to the products of *L. casei* and *L. plantarum*, arresting the regrowth to half of the carrying capacity of the untreated culture of *E. faecalis* at concentrations higher than 10%. The conditioned medium of a phylogenetically distinct probiotic bacterium, *B. coagulans*, was capable of inhibiting regrowth at concentrations higher than 20%, but could not eliminate regrowth altogether, indicating that the activity is a specific property of Lactobacillaceae (Figure 5A). The PBS control was not sufficient to prevent regrowth, indicating that the growth arrest is not due to starvation (Appendix A). Interestingly, planktonic *E. faecalis* cells were more resistant to conditioned media from all bacteria (Figure 5B); while a partial response (e.g., growth arrest at concentration >30%) was observed for Lactobacillaceae conditioned media, no biologically significant growth inhibition (>50% of the untreated carrying capacity) was observed for *B. coagulans.* These results place the use of Lactobacillaceae as a promising therapeutic strategy for *E. faecalis* mediated root canal infections, specifically targeting *E. faecalis* biofilms.

## 4. Discussion

One of the primary goals of root canal treatment is to eliminate bacteria from the root canal system in order to treat or prevent apical periodontitis [23]. However, biofilm cells are able to effectively evade the immune system and are more resistant to antimicrobial agents by several orders of magnitude compared with their planktonic counterparts [37]. Therefore, bacterial biofilms pose a major obstacle to endodontic disinfection in root canal systems [38]. Currently, the key element in the elimination of intra-canal biofilms is the use of anti-microbial irrigating solutions during root canal treatment [23].

Here, we assessed the effectiveness of the most commonly used anti-microbial irrigation solution, sodium hypochlorite, towards pre-established biofilms and found that sodium hypochlorite irrigations have a limited ability to eliminate the *E. faecalis* biofilms completely. Furthermore, while all working concentrations of sodium hypochlorite failed to prevent the regrowth of *E. faecalis* biofilms (Figure 3), irrigation of pre-established biofilms with 1% sodium hypochlorite enhanced the biofilm biomass over polystyrene surfaces (Figure 1 and Figure 2). This significant increase was reversible by DNase treatment and indicated that, similar to other bacterial biofilm models, the lysis of a subpopulation of biofilm cells contributes to the release of eDNA that can function as an adhesion [39,40] and, thereby, enhances the adhesion of bacteria to surfaces. These results highlighted the importance of sodium hypochlorite concentration as critical for the desired outcome of the irrigation. Importantly, we found that planktonic cells (as judged by the biofilm inhibition essay) are sensitive to sodium hypochlorite at all working concentrations (Figure 1) and cannot be used to assess the response of root-canal biofilms to antimicrobial substances.

Unlike antimicrobial substances, recent studies indicate that probiotics may be effective agents in the elimination of biofilms. Probiotic strains and their products can modify host mucosal and systemic immune responses and protect the host against pathogenic biofilms. *Lactobacillus* (Lactic Acid Bacteria, LAB) were suggested to potentially modulate the microbial ecology of biofilms by pathogen growth inhibition, adhesion, and co-aggregation and were shown to exert antimicrobial activities against the gastrointestinal (GI) tract pathogens via declining luminal pH, competing for adhesion sites and nutrients, and producing antimicrobial agents such as bacteriocins, hydrogen organic acids, and oxidizing agents [32].

Based on these properties, probiotics were suggested as a promising therapy for pathogenic biofilms in the GI [32]. However, as probiotics are considered safe to be administered orally (and acknowledged as GRAS (Generally Recognized As Safe) by the U.S. Food and Drug Administration), they are strong candidates to treat infections in the oral cavity [32].

Consistently, preliminary studies indicate that *Lactobacilli* (*Lactobacillcae* strains) can prevent the formation of biofilm in *S. mutans* associated with dental plaque, and *Lactococcus lactis* NCC2211, as a nonpathogenic dairy probiotic, was able to modify the growth of the cariogenic *Streptococcus sobrinus* OMZ176.53 [41]. The probiotic bacterium *Lactobacillus rhamnusos* was shown to inhibit the growth of planktonic *E. faecalis* cells [42], and lipoteichoic acid (Lp.LTA), a polymer from the microbial cell wall, purified from *L. plantarum* disrupted *E. faecalis* biofilm formation but did not efficiently kill the biofilm cells [43]. Thus, following our findings that sodium hypochlorite irrigations fail to treat established *E. faecalis* biofilms (Figure 1, Figure 2 and Figure 3), we evaluated the products of several probiotic strains belonging to the *Firmicutes* phylum as potential substitutes/additives to treat root canal infections, focusing on their potential role as additives/substitutes for sodium hypochlorite irrigations.

Our results indicated that the secreted products of probiotic LAB strains (*L. casei* and *L. plantarum*), but not of an alternate probiotic bacterium belonging to the same phylum (*B. coagulans*), are efficient agents capable of fully abolishing cells from pre-established biofilms and effectively preventing their regrowth (Figure 4 and Figure 5). Unlike sodium hypochlorite treatment that was more efficient for planktonic cells, the formation of biofilms sensitizes *E. faecalis* cells for probiotic secretions, allowing their efficient eradication in concentrations inert to planktonic bacteria (Figure 5). This result is not intuitive, as biofilm cells are generally considered more resistant to antimicrobials [44], although similar resistance to biofilm and planktonic cells of the opportunistic pathogen *Pseudomonas aeruginosa* has also been detected [45]. Notably, biofilm cells of *Bacillis subtilis* are more sensitive than their planktonic counterparts to agents that target the cell wall [46] which may suggest that the secretome of LAB strains includes antimicrobial compounds that target the cell envelope. While the exact nature of the antimicrobial compounds that preferentially target biofilms remains to be determined, their existence in the repertoire of bioactive molecules produced by probiotic strains cannot be overestimated.

## 5. Conclusions

Sodium hypochlorite irrigations may contribute to the persistence of biofilm cells if used at concentrations lower than 3%. In contrast, probiotic strains and their products represent a new reservoir of biofilm therapies to treat the pre-established biofilms of *Enterococcus faecalis* infections formed in the root canal system.

From our studies, it can be concluded that the secreted products of probiotic *lactobacillus* strain’s growth can efficiently kill biofilm cells and prevent their regrowth, while the toxic irrigation solution of sodium hypochlorite should be used with caution. Our results strongly suggest that probiotic strains or their products represent an innovative, safe therapeutic tool for biofilm-related root canal infection. However, further research and reliable in vivo studies are needed for transferring this treatment strategy to human subjects.

## Figures and Tables

**Figure 1 pharmaceutics-14-00751-f001:**
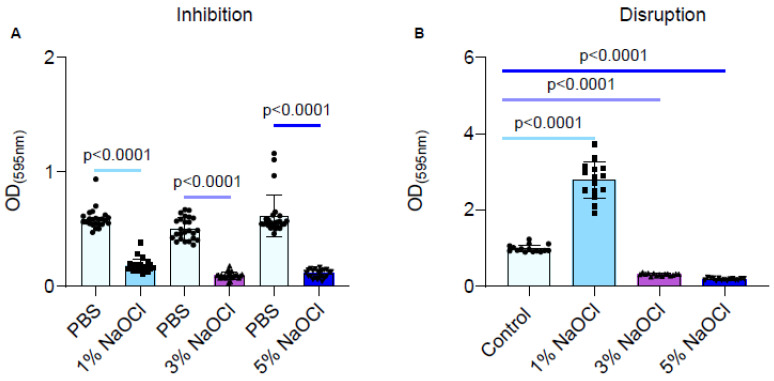
Different concentrations of NaOCl inhibited the biofilm formation of *E. faecalis* but failed to eradicate pre-established biofilms. Each color represents a different NaOCl concentration (**A**) *E. faecalis* cells were diluted 1:100 into a fresh TSB medium (untreated) or applied with different concentrations of NaOCl (1%, 3% or 5%). 100 μL of cultures were split into a 96-well polystyrene plate and further incubated at 37 °C for 24 h. Biofilm formation was assessed by crystal violet staining. (**B**) *E. faecalis* cells were diluted 1:100 into a fresh medium. 100 μL of cultures were split into a 96-well polystyrene plate and further incubated at 37 °C for 24 h. The growth media was removed, and the remaining biofilm biomass was either untreated or treated with different concentrations of NaOCl (1%, 3%, or 5%) for 5 min. Biofilm was assessed by crystal violet staining. Graphs represent the mean ± SD from five biological repeats (*n* = 25). Statistical analysis was performed using Brown–Forsthye and Welch’s ANOVA with Dunnett’s T3 multiple comparisons test. *p* < 0.05 was considered statistically significant.

**Figure 2 pharmaceutics-14-00751-f002:**
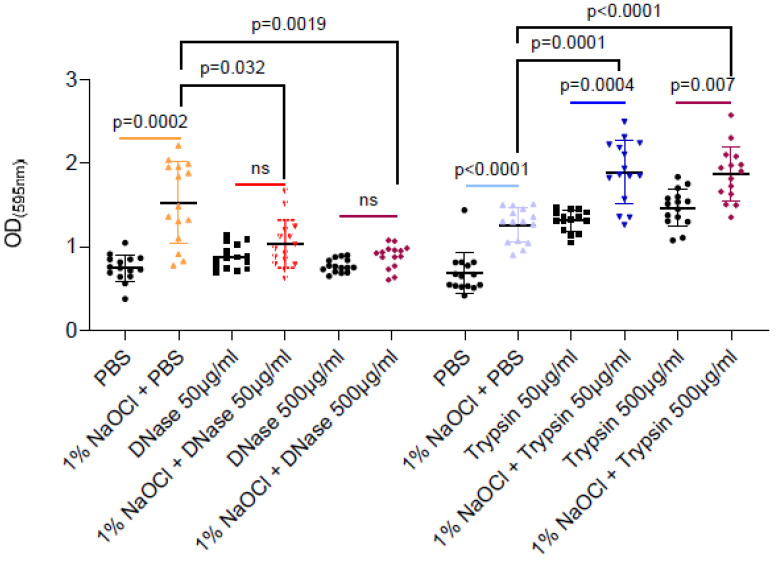
DNase but not trypsin treatment efficiently reduced the remaining biofilm biomass- treatments are indicated in the vertical axis’ legend. *E. faecalis* cells were diluted 1:100 into a TSB fresh medium. 100 μL of cultures were split into a 96-well polystyrene plate and further incubated at 37 °C for 24 h. Biofilm was untreated or treated with different concentrations of NaOCl (1%, 3% or 5%) for 5 min. Biofilm cells were washed with PBS, and 100 μL of PBS/DNase/Trypsin in the indicated concentrations were added to each well and further incubated at 37 °C for 24 h. Biofilm biomass was assessed by crystal violet staining. Graphs represent the mean ± SD from five biological repeats (*n* = 25). Statistical analysis was performed using Brown–Forsthye and Welch’s ANOVA with Dunnett’s T3 multiple comparisons test. *p* < 0.05 was considered statistically significant.

**Figure 3 pharmaceutics-14-00751-f003:**
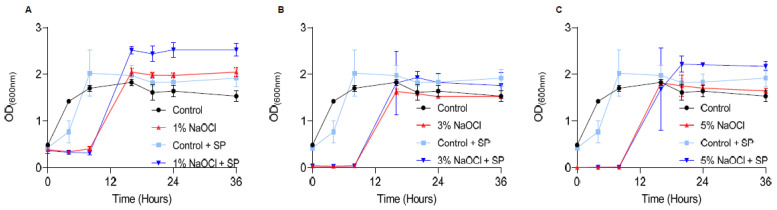
Sodium hypochlorite irrigation results in the regrowth of biofilm cells. *E. faecalis* cells were diluted 1:25 into a fresh TSB medium. 1000 μL of cultures were split into a 12-well polystyrene plate and further incubated at 37 °C for 24 h. Biofilm was untreated or treated with different concentrations of NaOCl (1% (**A**), 3% (**B**), or 5% (**C**)) for 5 min. Bacterial cells were collected with 200 μL PBS and added to 3 mL of TSB medium applied with 0.5% glucose either applied or not with sodium pyruvate (2 mg/mL) and regrown at 37 ℃ with shaking. OD600 was measured at different time points (0 h, 4 h, 8 h, 16 h, 20 h, 24 h and 36 h). Graphs represent the mean ± SD from three independent experiments (*n* = 9). Statistical analysis was performed using two-way ANOVA followed by Dunnett’s multiple comparison test. *p* < 0.05 was considered statistically significant.

**Figure 4 pharmaceutics-14-00751-f004:**
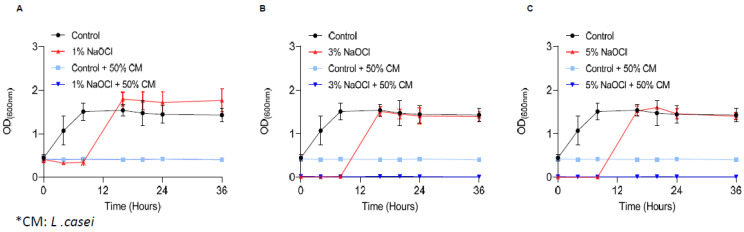
Secreted products of *L. casei* prevent the regrowth of *E. Faecalis* biofilm cells. *E. faecalis* cells were diluted 1:25 into a fresh TSB medium. 1000 μL of cultures were split into a 12-well polystyrene plate and further incubated at 37 °C for 24 h. Biofilm was untreated or treated with different concentrations of NaOCl (1% (**A**), 3% (**B**), or 5% (**C**)) for 5 min. Bacterial cells were collected with 200 μL PBS and added to 3 mL of TSB medium (applied with 0.5% glucose) either mixed 1:1 with conditioned medium of *L. casei* or not. Cells were regrown at 37 ℃ with shaking. OD600 was measured at different time points (0 h, 4 h, 8 h, 16 h, 20 h, 24 h and 36 h) as described above. Graphs represent the mean ± SD from three independent experiments (*n* = 9). Statistical analysis was performed using two-way ANOVA followed by Dunnett’s multiple comparison test. *p* < 0.05 was considered statistically significant.

**Figure 5 pharmaceutics-14-00751-f005:**
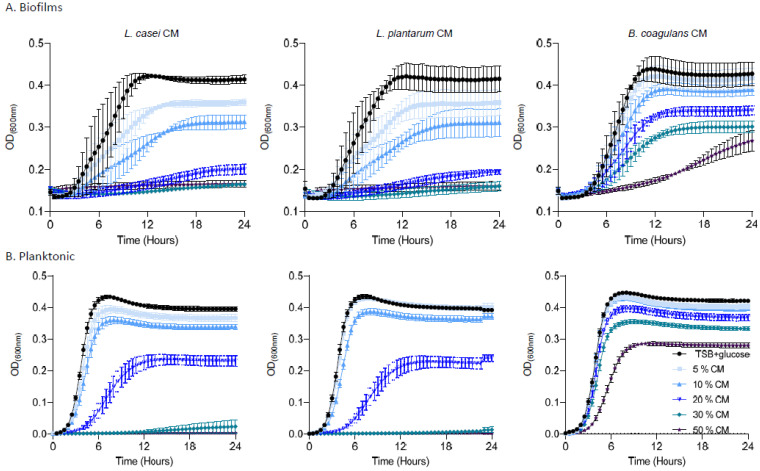
The effect of the secreted products of probiotic strains on *E. Faecalis* biofilm and planktonic cells. (**A**) Growth of *E. faecalis* biofilms applied with different concentrations (5%, 10%, 20%, 30%, or 50%) of conditioned medium from either *L. casei*, *L. plantarum*, or *B. coagulans*. Different colors represent the indicated concentrations of the CM. (**B**) Growth of *E. faecalis* planktonic cells applied with different concentrations (5%,10%, 20%, 30%, or 50%) of conditioned medium from *L. casei*, *L. plantarum*, or *B. coagulans*. Prevention of growth was accomplished by a high concentration of *L. casei*/*L. plantarum* with a dose dependent growth reduction. The *B. coagulans* conditioned medium failed to prevent *E. faecalis* growth. Graphs represent the mean ± SD from three independent experiments (*n* = 9). Statistical analysis was performed using two-way ANOVA followed by Dunnett’s multiple comparison test. *p* < 0.05 was considered statistically significant.

## Data Availability

All relevant data sets, materials and methods are available in full within the manuscript.

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
