# Peer review of "The Products of Probiotic Bacteria Effectively Treat Persistent Enterococcus faecalis Biofilms"

_pharmaceutics, 2022, doi:10.3390/pharmaceutics14040751_

Round 1
Reviewer 1 Report
The paper entitled “The Products of Probiotic bacteria Effectively Treat Persistent Enterococcus faecalis Biofilms” focuses on the effectiveness of probiotic strains to prevent the regrowth of Enterococcus faecalis biofilm cells treated by sodium hypochlorite irrigation. Both strains Lactobacillus casei and Lactobacillus plantarum were found to be capable of fully eradicating established infections and preventing re-infection following the root canal treatment and thus to be used as a feasible therapeutic approach for these persistent infections.
Тhe introduction refers to the aim of the study, the experimental part is consistently revealed and explained while the results are understandably submitted and well-illustrated. The conclusion summarizes the aforementioned results. In my opinion, the paper should be interesting from a scientific and practical point of view.
I would like to recommend the publication of paper publication after some changes concerning the following issues:
- More quantitative findings can be included in the abstract;
- Some abbreviations such as “SP”, “CM”, “ns”, etc, need to be clarified in the text;
- It is not clear from the text, what constitutes the “novel dentin experimental model”.
- The finding that “… planktonic E. faecalis cells were more resistant to conditioned media from all bacteria” is worth discussing in more detail in the Discussion section.
- In order to reveal the biochemical character of the inhibiting substances in the conditioned medium of L. casei, L. plantarum, and B. coagulans, additional analysis or, at least, references for that should be given.
- Where appropriate, the results can be compared with similar studies.
- Additional information can be added to the conclusion section.
Author Response
Dear reviewers and editors,
We thank the reviewers for taking the time to critically read our manuscript. We believe that the revised manuscript addresses their remaining concerns in full.
#Reviewer 1
- More quantitative findings can be included in the abstract.
We thank the reviewer for the comment; we have now elaborated on the abstract section.
- Some abbreviations such as “SP”, “CM”, “ns”, etc, need to be clarified in the text
We have now updated the abbreviations.
- It is not clear from the text, what constitutes the “novel dentin experimental model”.
this sentence is now revised and the word “novel” was removed
- The finding that “… planktonic E. faecalis cells were more resistant to conditioned media from all bacteria” is worth discussing in more detail in the Discussion section.
We now address this point with elaboration as follows: “The formation of biofilms sensitizes E. faecalis cells for probiotic secretions allowing their efficient eradication in concentrations inert to planktonic bacteria (Figure 5). This result is not intuitive as biofilm cells are generally considered more resistant to antimicrobials3, although similar resistance of biofilm and planktonic cells of the opportunistic pathogen P. aeruginosa was also detected4. Notably, biofilm cells of Bacillis subtilis are more sensitive than their planktonic counterparts to agents that target the cell wall 5, which may suggest that the secretome of LAB strains includes antimicrobial compounds that target the cell envelope.”
- In order to reveal the biochemical character of the inhibiting substances in the conditioned medium of L. casei, L. plantarum, and B. coagulans, additional analysis or, at least, references for that should be given.
We thank the reviewer for the comment. We now explained the possible inhibiting sources in the CM of these strains, with appropriate references. “These strains produce an extremely broad repertoire of antimicrobial organic acids, bacteriocins, and enzymes which were reviewed extensively 1.” We note that a few dozens of natural products could account for this biofilm specific effect. We now aim to purify them in subsequent studies.
- Where appropriate, the results can be compared with similar studies.
We have now added references to relevant studies.
- Additional information can be added to the conclusion section.
We thank the reviewer for the comment, and expanded our conclusion section, in addition, we elaborated more on the points above in the discussion (notably, the journal's format requires both a discussion and conclusion sections).

Reviewer 2 Report
In the article (pharmaceutics-1614763), the authors reported the efficacy of probiotic bacteria and their products on the inhibition and treatment of Enterococcus faecalis. The authors showed that sodium hypochlorite at concentrations upto 5% is not able to fully inhibit the growth of the biofilm. Toward these results, the authors have evaluated the ability of different probiotic bacteria media to treat and inhibit the biofilm growth. The results showed that different probiotic bacteria strains were able to dose-dependently inhibit the biofilm growth. Interestingly, the authors showed that the probiotic bacteria and their products were not able to fully inhibit or treat Enterococcus faecalis. However, the full inhibition was obtained after synergistic treatment with at least 3% of sodium hypochlorite. Overall, this is quite interesting study. However, this study lack of novelty. Further, the exact mechanism of the inhibitory activity has not addressed in this study. I would not suggest the publication of this study in the present form. The following are the detailed concerns:
- this study is lacking of novelty. Several studies have reported the inhibitory potency of probiotic bacteria toward biofilm formation. The authors should present more details about the mode of action for the observed inhibitory activity.
- the authors should also investigate what the exact products produced by probiotic bacteria strain that led to the inhibitory activity. As presented, it does not provide any kind of information. The authors showed present the exact content of the medium produced by the different probiotic bacteria strains and accordingly they can offer this content as format for further therapeutic investigations.
- the authors should investigate the effect of the different probiotic bacteria strains at the presented concentrations on the different cell lines, what about the cytotoxicity??
- the results part is really poorly presented. The authors should divide it into subsections with subtitle.
- the authors stated in abstract and through discussion/conclusion that ''Sodium hypochlorite irrigation may contribute to the persistence of biofilm cells''. Based on the presented results, I do not agree with the authors. Its clear from the results that the tested probiotic bacteria strains alone were not able to prevent the biofilm growth, and that sodium hypochlorite was needed for fully inhibition. It does not mean that 1% of sodium hypochloride was not able to fully inhibit biofilm growth, that sodium hypochlorite irrigation may contribute to the persistence of biofilm cells. How??!!!
- To state that the presented findings can lead to in vivo investigation is over estimations and the authors should change this in the conclusion part.
- Finally, the manuscript requires extensive language editing for academic style.
Author Response
In the article (pharmaceutics-1614763), the authors reported the efficacy of probiotic bacteria and their products on the inhibition and treatment of Enterococcus faecalis. The authors showed that sodium hypochlorite at concentrations upto 5% is not able to fully inhibit the growth of the biofilm. Toward these results, the authors have evaluated the ability of different probiotic bacteria media to treat and inhibit the biofilm growth. The results showed that different probiotic bacteria strains were able to dose-dependently inhibit the biofilm growth. Interestingly, the authors showed that the probiotic bacteria and their products were not able to fully inhibit or treat Enterococcus faecalis. However, the full inhibition was obtained after synergistic treatment with at least 3% of sodium hypochlorite. Overall, this is quite interesting study. However, this study lack of novelty. Further, the exact mechanism of the inhibitory activity has not addressed in this study. I would not suggest the publication of this study in the present form. The following are the detailed concerns:
We thank the reviewer for the comment. Actually, we did not ask whether probiotic strains could inhibit biofilm growth in Enterococcus faecalis but rather whether their products can eradicate in full and kill the resident cells of pre-established biofilms. The most common cause of recurring endodontic infections. Previous studies have shown that probiotics can inhibit biofilm formation, however, we did not try to inhibit biofilm formation as it is not relevant for root canal infections, where pre-established biofilms are treated by different agents as done in our setting. Our work is focused on the eradication of E. faecalis cells from pre-established biofilms, and to the best of our knowledge, this is the first time where these particular settings were directly tested with probiotic bacteria, although they are highly relevant for infections. We now clarified the research question in sections' titles.
Furthermore, we obtained evidence for eDNA dependent increase in the biofilm biomass following NaOCl treatment. This treatment is currently routinely used in oral medicine, and the potential of deleterious effects from the wrong working concentration of sodium hypochlorite cannot be overestimated
Finally, we systematically explored the different probiotic strains on for treating established biofilms. As the reviewer has pointed out in the next remarks, the exact mechanism on how the molecules from the probiotic strains inhibit E. faecalis biofilm formation, the exact chemical nature of these molecules will require further investigation. We now acknowledge this point in our revised discussion
This study is lacking of novelty. Several studies have reported the inhibitory potency of probiotic bacteria toward biofilm formation. The authors should present more details about the mode of action for the observed inhibitory activity.
We thank the reviewer for the comment. We ask you to consider our response above: “Actually, we did not ask whether probiotic strains could inhibit biofilm growth in Enterococcus faecalis but rather whether their products can eradicate in full and kill the resident cells of pre-established biofilms. The most common cause of recurring endodontic infections. Previous studies have shown that probiotics can inhibit biofilm formation, however, we did not try to inhibit biofilm formation as it is not relevant for root canal infections, where pre-established biofilms are treated by different agents as done in our setting. Our work is focused on the eradication of E. faecalis cells from pre-established biofilms, and to the best of our knowledge, this is the first time where these particular settings were directly tested with probiotic bacteria, although they are highly relevant for infections. We now clarified the research question in sections' titles.
Furthermore, we obtained evidence for eDNA dependent increase in the biofilm biomass following NaOCl treatment. This treatment is currently routinely used in oral medicine, and the potential of deleterious effects from the wrong working concentration of sodium hypochlorite cannot be overestimated.
Finally, we systematically explored the different probiotic strains on for treating established biofilms. As the reviewer has pointed out in the next remarks, the exact mechanism on how the molecules from the probiotic strains inhibit E. faecalis biofilm formation, the exact chemical nature of these molecules will require further investigation. We now acknowledge this point in our revised discussion”
The authors should also investigate what the exact products produced by probiotic bacteria strain that led to the inhibitory activity. As presented, it does not provide any kind of information. The authors showed present the exact content of the medium produced by the different probiotic bacteria strains and accordingly they can offer this content as format for further therapeutic investigations.
We thank the reviewer for the comment; we have now explained in our discussion section various agents that might be responsible for this inhibitory activity. Also, in line with reviewer one comment.
The authors should investigate the effect of the different probiotic bacteria strains at the presented concentrations on the different cell lines, what about the cytotoxicity?
We thank the reviewer for the comment. As of now, all bioactive probiotic strains and their products are consumed at similar or even higher concentrations during the daily consumption of probiotic products by a significant portion of the population. All the strains are all considered as GRAS by the FDA (https://www.fda.gov/food/food-ingredients-packaging/generally-recognized-safe-gras). Thereof, testing toxicity seem a bit redundant with previous FDA approved studies
The results part is really poorly presented. The authors should divide it into subsections with subtitle.
We now divided the results into sections.
The authors stated in abstract and through discussion/conclusion that ''Sodium hypochlorite irrigation may contribute to the persistence of biofilm cells''. Based on the presented results, I do not agree with the authors. Its clear from the results that the tested probiotic bacteria strains alone were not able to prevent the biofilm growth, and that sodium hypochlorite was needed for fully inhibition. It does not mean that 1% of sodium hypochlorite was not able to fully inhibit biofilm growth, that sodium hypochlorite irrigation may contribute to the persistence of biofilm cells. How?
As we explain and demonstrate, low concentration of sodium hypochlorite incrase biofilm adherence, probably by the release of Edna.
To state that the presented findings can lead to in vivo investigation is over estimations and the authors should change this in the conclusion part.
We have now expanded and edited our conclusion.
Finally, the manuscript requires extensive language editing for academic style.
We have revised the manuscript with an English editor.

Round 2
Reviewer 1 Report
The authors have carefully addressed all the review’s recommendations and the manuscript has been substantially improved.
Author Response
We thank the reviewer for the comments and for the positive evaluation of our work. In addition, we sent the manuscript for a professional English editor, and we believe that the editorial revision successfully addressed the remaining concerns below of the reviewer.
Reviewer 2 Report
The authors have adequately responded to all concerns that have been raised in the first report. However, there are few points that still require to be addressed:
- Regarding the exact products produced by probiotic bacteria strain that led to the inhibitory activity and the possibility of their cytotoxicity, the authors replied that they have explained in the discussion section various agents that might be responsible for this inhibitory activity. However, I did not see that the authors have discussed this in the discussion section. Accordingly, this part still missing.
- Also, the authors mentioned that ''products produced by probiotic bacteria strain are consumed at similar or even higher concentrations during the daily consumption of probiotic products by a significant portion of the population. All the strains are all considered as GRAS by the FDA (https://www.fda.gov/food/food-ingredients-packaging/generally-recognized-safe-gras).''. Please specify more which products you are talking about? and why this is not mentioned or discussed in the discussion part?
- the manuscript has not been edited for language, grammatical mistakes and academic style.
Author Response
Point 1: "The authors have adequately responded to all concerns that have been raised in the first report. However, there are few points that still require to be addressed". We thank the reviewer for the evaluation of the efforts invested during the revision to address the previous concerns of the reviewer
- Regarding the exact products produced by probiotic bacteria strain that led to the inhibitory activity and the possibility of their cytotoxicity, the authors replied that they have explained in the discussion section various agents that might be responsible for this inhibitory activity. However, I did not see that the authors have discussed this in the discussion section. Accordingly, this part still missing.
We have addressed the issue in length in our introduction as following "Probiotic microorganisms are living microorganisms that are capable of providing various health benefits to the host including the elimination of microbial pathogens. Therefore, delivery of probiotics may eradicate or diminish more pathogenic bacteria32, but is not currently used to treat root canal infections effectively. Species of the family Lactobacillaceae and Bacillus that are naturally found in the human gastrointestinal (GI) tract are often considered models of beneficial microbiota. These strains produce an extremely broad repertoire of antimicrobial organic acids, bacteriocins, and enzymes which were reviewed extensively32. " We do not see the benefit of repeating the same idea in the discussion. Importantly, as the strains and their products as whole are consumed in yogurts, and as probiotics but not high concentrations of a specific product, we are not extending at this stage our conclusions to specific products.
We also acknowledge this in the discussion as following "While the exact nature of the antimicrobial compounds that preferentially target biofilms remains to be determined, their existence in the repertoire of bioactive molecules produced by probiotic strains cannot be overestimated."
"
- Also, the authors mentioned that ''products produced by probiotic bacteria strain are consumed at similar or even higher concentrations during the daily consumption of probiotic products by a significant portion of the population. All the strains are all considered as GRAS by the FDA (https://www.fda.gov/food/food-ingredients-packaging/generally-recognized-safe-gras).''. Please specify more which products you are talking about? and why this is not mentioned or discussed in the discussion part?
Currently these probiotic strains and their products are frequently consumed with probiotic yogurts (as well as fermented food) but also as purified dissected strains. We chose not to extend on this aspect as it seem a bit out of scope for this work. However, we now refer to this aspect in the discussion directing the readers to an elaborated review from IKG [Suissa et al., Trends in Microbiology, 2022] on the topic: "Based on these properties, probiotics were suggested as a promising therapy for pathogenic biofilms in the GI32. However, as probiotics are considered safe to be administered orally (and acknowledged as GRAS (Generally Recognized As Safe) by the U.S. Food and Drug Administration), they are strong candidates to treat infections in the oral cavity32.
"
- The manuscript was not edited for language, grammatical mistakes and academic style. The manuscript was sent to a professional English editor proofing it to adhere to the US English language code